# EA²N: Evidence-based AMR Attention Network for Fake News Detection

## Abstract

Proliferation of fake news has become a critical issue in today's information-driven society. Our study includes external knowledge from Wikidata and deviates from the reliance on social information to detect fake news, that many state-of-the-art (SOTA) fact-checking models adopt. This paper introduces **EA²N**, an **E**vidence-based **AMR A**ttention **N**etwork for Fake News Detection. EA²N leverages Abstract Meaning Representation (AMR) and incorporates knowledge from Wikidata using proposed evidence linking algorithm, pushing the boundaries of fake news detection. The proposed framework encompasses a combination of novel language encoder and graph encoder to detect the fake news. While the language encoder effectively combines transformer encoded textual features with affective lexical features, the graph encoder encodes AMR with evidence through external knowledge, referred as WikiAMR graph. A path-aware graph learning module is designed to capture crucial semantic relationships among entities over evidence. Extensive experiments supports our model's superior performance, surpassing SOTA methodologies. This research not only advances the field of Fake News Detection but also showcases the potential of AMR and external knowledge for robust NLP applications, promising a more trustworthy information landscape.[1]

## 1 Introduction

Social media has revolutionized the exchange of information by enabling people to obtain and share news online. However, with the growing popularity and convenience of social media, the dissemination of fake news has also escalated. The deliberate distortion and fabrication of facts in fake news have severe negative consequences for individuals and society (Brewer et al., 2013). Therefore, it is crucial and socially advantageous to detect and address fake news on social media.

Significant efforts have been made in the direction of fake news detection in the past decade. Early works (Feng et al., 2012; Ma et al., 2016) used manually crafted textual features to detect fake news. Later, many researchers used LSTM and RNN based methods (Long et al., 2017; Liu & Wu, 2018) for the purpose. Deep learning methods used therein is able to learn text features out of the article. Recently, external knowledge is incorporated alongside the textual features to improve fake news detection models. Dun et al. (2021) proposed KAN model which leverages external evidence from the Wikidata. On the contrary, FinerFact (Jin et al., 2022) and Dual-CAN (Yang et al., 2023), incorporates social information that supports authenticity of the news article. Despite the significant achievements, these methods exhibit

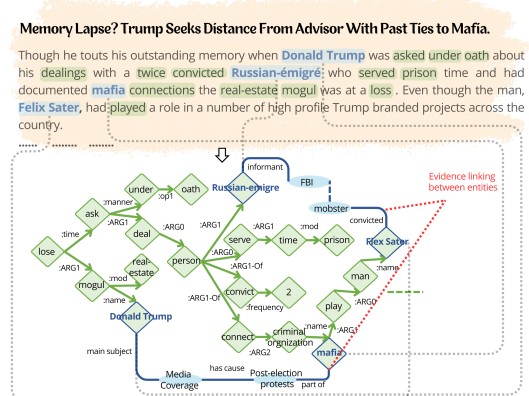

Figure 1: An example of evidence linking in AMR graph constructed over Politifact news article.

---

[1]Code repository: Code will be shared once the discussion forum start.

limitations in capturing certain details. Precisely, these models struggle to maintain longer text dependencies and less effective to capture complex semantic relations such as events, locations, trigger words and so on. Additionally, the way of incorporating external knowledge into these models is not highly reliable and time consuming. For example, KAN only considers single entity contexts and fail to link context between two entities. On the other hand, FinerFact gather supported claims from social platforms that is time-consuming. Although social authenticity produces good results, these information can be manipulated by social media users for personal gain. In order to tackle these challenges, our study effectively uses complex semantic relations of news articles and evidence found in Wikidata (Wikidata5M (Wang et al., 2021a)) with the help of a novel graph representation.

In this paper, we present a novel model for detecting fake news that leverages a semantically enriched knowledge base to classify a news article as real or fake. Our model incorporates Abstract Meaning Representation (AMR) (Banarescu et al., 2013) to understand the logical structure of sentences. Further, the model establishes relations between entities found in AMR graph, through a new evidence linking algorithm. The algorithm utilizes Wikidata to connect evidence, leading to the formation of a graph referred as WikiAMR. To the best of our knowledge, this is the first study to examine the evidence based semantics features of AMR graphs for fake news detection. For the illustration, Fig. 1 shows an example of WikiAMR constructed over a news article using Wikidata. In this, a linkage is established between the entities '$Donald\ Trump$' and '$Mafia$', wherein the nodes '$Media\ Coverage$' and '$Post-election\ protest$' are connected by relations such as '$main\ subject$', '$has\ cause$', and '$part\ of$' in the path. Similarly, a connecting evidence path emerges between '$Russian-emigre$' and '$Flex\ Ster$'. These instances serve as valuable evidence to assess the credibility of news content. Next, to encode the WikiAMR graph, we employ a path-aware graph learning module. This module uses relation-enhanced global attention that focus on important relation over entities and compute the attention score considering the entities and their relations. By modifying the Graph Transformer (Cai & Lam, 2020) for entities, our model can effectively reason over the relation paths within the WikiAMR graph. In order to enhance the capabilities of language encoder, we also use affective features (Ghanem et al., 2021) extracted from different segments of news article and concatenated with language embedding. Finally, the representations of language and AMR graph are fed into a classification layer using transformer to predict the veracity of the news. The key contributions of our research are as follows:

- Introduction of EA$^2$N, a novel Evidence-based AMR Attention Network for Fake News Detection, reasoning over evidence linked through external knowledge.
- Introduction of WikiAMR graph, a novel graph structure that includes undirected evidence paths, extracted form external knowledge graph, between entities of AMR constructed from text document.
- Evidence Linking Algorithm to generate WikiAMR, from entity-level and context-level filtering to enhance model performance.
- Comprehensive evaluation of EA$^2$N against state-of-the-art techniques, demonstrating its superior performance and effectiveness.

## 2 RELATED WORKS

In this section, we delve into brief of the approaches employed in the detection of fake news. We have categorized the relevant studies into three components: textual-based methods, knowledge-aware methods, and AMR-based methods. A comprehensive explanation of each approach is provided in the subsequent sections.

**Textual-based** approaches primarily rely on the textual content extracted from articles to verify the authenticity of news. In the early times, the emphasis was primarily on developing a supplementary collection of manually created features rooted in linguistic characteristics (Feng et al., 2012; Ma et al., 2016; Long et al., 2017; Rashkin et al., 2017; Liu & Wu, 2018). These early studies demanded extensive efforts to assess the efficacy of these manually crafted features. Recently, Ghanem et al. (2021) proposed FakeFlow that involves the utilization of a text with lexical features to classify news as fake news. Early detection of fake news is facilitated in many work (Wei et al., 2021; Azevedo et al., 2021), however, their effectiveness is limited as they overlook auxiliary knowledge that could aid in news verification.

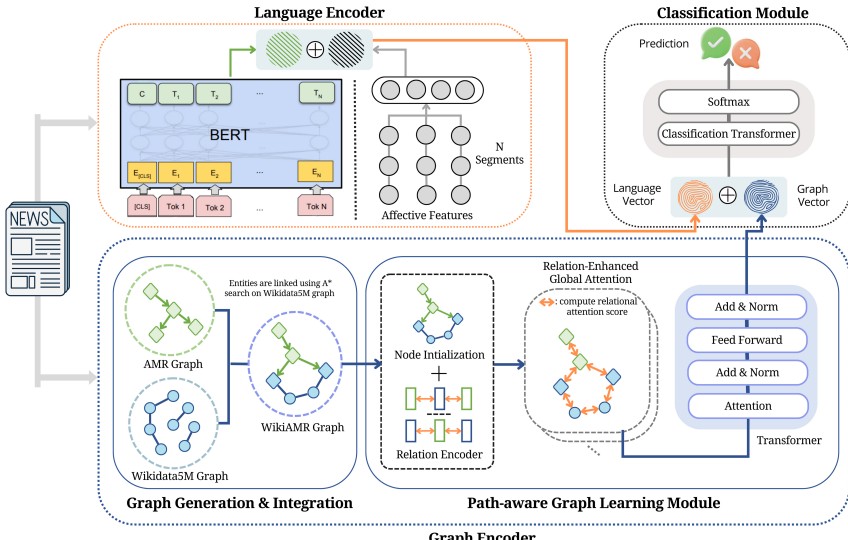

Figure 2: EA$^2$N framework for fake news detection.

**Knowledge-aware** methods utilize auxiliary knowledge to aid in the process of news verification. Monti et al. (2019) extends classical CNNs to operate on graphs by analyzing the user activity, content, social graph etc., while Shu et al. (2020b) found user profile features to be useful in fake news detection. Lu & Li (2020) employ a co-attention model that uses both news content and social context. Later, Wang et al. (2020) and Hu et al. (2021) utilize entity linking to capture entity descriptions from Wikidata and integrate them into their models for the identification of fake news. Recently, KAN (Dun et al., 2021) considered external knowledge from Wikidata to expand domain knowledge and Jin et al. (2022) presented Finerfact, a fine-grained reasoning framework using social information to detect the fake news. Most recently introduced Dual-CAN (Yang et al., 2023) method takes news content as well as social media replies, and external knowledge for the purpose. Thus it is proven that better fake news detection require external knowledge acquisition.

**AMR** as introduced by Banarescu et al. (2013), represents relations between nodes using Prop-Bank, frameset and sentence vocabularies. It utilizes over hundred semantic relations, including negation, conjunction, command, and wikification. It aims to represent different sentences with the same semantic meaning using the same AMR graph. Various NLP fields, such as summarization (Kouris et al., 2022), event detection (Wang et al., 2021b), question answering (Lim et al., 2020) etc., have effectively used AMR. Despite its wide range of applications in NLP, AMR has not been investigated to capture complex semantic relations in documents for fake news detection. Recently, (Zhang et al., 2023) used AMR for identification of out-of-context multimodal misinformation. Understanding the importance of semantic relations, we embarked on an exploration of utilizing AMR for the purpose of detecting fake news.

## 3 PROPOSED METHOD

We present a brief overview of our proposed model, Evidence-based AMR Attention Network (EA$^2$N), depicted in Fig. 2. Our framework comprises three key components: Language Encoder, Graph Encoder, and Classification Module. Description for each component is provided in the following sections. Before proceeding further, let us define the problem statement formally.

### 3.1 PROBLEM STATEMENT

Our objective is to perform binary classification on news articles, classifying them as either real ($y = 0$) or fake ($y = 1$). Formally, given a news article $S$, the task is to learn a function $F$ such that $F : F(S) \rightarrow y$, where $y \in \{0, 1\}$ represents the ground truth labels of the news articles.

## 3.2 LANGUAGE ENCODER

We utilized the BERT (Bidirectional Encoder Representations from Transformers) (Devlin et al., 2019) model to encode sequences of tokens in sentences. Let a news article $S$ be denoted as tuple$(T, B)$, where $T$ represents the title and $B$ represents the body text of the news article, both containing a set of words $\{w_1, w_2, \ldots, w_n\}$. In order to create the final text for input, we concatenate the title and body using a special "[SEP]" tag. After that, we initially tokenize these words and fed into the BERT model to obtain the final layer embedding $\mathrm{H}^i$ in the following manner:

$$S = Concat([T, B]), [\mathrm{H}_S^0, \mathrm{H}_S^1, \ldots, \mathrm{H}_S^{N_S}] = \mathrm{BERT}(S), \mathcal{Z}^S = \mathrm{H}_S^0 \tag{1}$$

We extracted affective lexical features (Ghanem et al., 2021) such as emotions, sentiment, morality, hyperbolic, and imagebility from the $n$ different segments $\{s_1, s_2, \ldots, s_n\}$ of article $S$ to enhance the capabilities of the language encoder. This enhance the capabilities to differentiate documents more effectively by capturing the distribution of the features segment wise. Each feature vector, $a_{s_j}$, of $j$-th segment $s_j \in S$ is represented with a term frequency that takes into account the articles' length as a weighting factor. This approach allows us to effectively capture and represent the distinctive characteristics of each segment in the article, accounting for variations in segment lengths. The resulting language vector, denoted as $\mathcal{Z}^l$, is obtained by concatenating the vector embedding derived from BERT with the representation vector formed by integrating the affective features as:

$$\mathcal{Z}^{affect} = Concat([a_{s_1}, a_{s_2}, \ldots a_{s_n}]) \tag{2}$$

$$\mathcal{Z}^l = Concat([\mathcal{Z}^S, \mathcal{Z}^{affect}]) \in \mathbb{R}^{N \times L \times D} \tag{3}$$

Here $N$ is the batch size, $L$ is the maximum sequence length, and $D$ is the dimension of the lanuage feature vector.

## 3.3 GRAPH ENCODER

The Graph Encoder module plays a crucial role in transforming the language representation into a structured and abstract form. We adopt Abstract Meaning Representation (AMR) (Banarescu et al., 2013) to encode the meaning of the news article in a graph structure. The graph encoder comprises three key components, each responsible for specific tasks in the process, ranging from the generation of AMR to the integration of external knowledge and, finally, the path-aware graph learning module. Before describing the concise overview of each component in the following subsections, let us define the notation of graph here. A graph G in general is represented with a tuple (V, E) where V is a set of node entities, and E represents relation edges. For different graphs used in our article, we used different notations for G= (V, E) without detailing out there.

### 3.3.1 AMR GENERATION

The generation process converts the news articles into a network of nodes and edges, capturing the relationships between different entities. AMR generation process involves parsing the news articles to extract linguistic information, including semantic roles, relations, and core events. For a news article $\mathcal{S}$, we represent the AMR graph as $\mathcal{G}^{amr} = (\mathcal{V}^{amr}, \mathcal{E}^{amr})$.

$(s \: / \: seek - 01$
$\quad : arg0 \: (p \: / \: person$
$\quad\quad : name \: (n \: / \: name : op1 \: "Donald \: Trumph"))$
$\quad : arg1 \: (d \: / \: distance - 01$
$\quad\quad : arg1 \: p$
$\quad\quad : arg2 \: (p2 \: / \: person$
$\quad\quad\quad : arg0 - of \: (a \: / \: advise - 01)$
$\quad\quad\quad : arg1 - of \: (t \: / \: tie - 01)$
$\quad\quad\quad\quad : name \: (n2 \: / \: name \: : op1 \: "mafia")$
$\quad\quad\quad\quad : time \: (p3 \: / \: past)))$

As an illustrative example, consider the sentence: "*Donald Trumph seeks distance from advisor with past ties to mafia.*" The corresponding AMR graph is presented in right side of section. The AMR graph is a directed acyclic graph that represents a hierarchical structure with nodes denoting entities ($Donald \: Trumph$, $mafia$, $seek$, etc). Edges ($arg0, arg1, name$, etc) capture the relationships between these entities, forming a semantically structured representation of $\mathcal{S}$.

### 3.3.2 EVIDENCE INTEGRATION WITH AMR

We propose an evidence linking algorithm to extract evidence rich paths from external knowledge graph among entities in AMR. Given $\mathcal{G}^{wiki} = (\mathcal{V}^{wiki}, \mathcal{E}^{wiki})$, a Wikidata knowledge graph, the

algorithm integrates $\mathcal{G}^{wiki}$ with AMR by entity-level filtering (ELF) and context-level filtering (CLF). ELF assesses the relevance between entities using a $Relatedness(\cdot)$ function. If the relevance exceeds the ELF threshold ($\gamma$), it initiates the CLF process to link evidence between entities.

**Entity-Level Filtering (ELF)**: During ELF, pairs of entities within the AMR graph are examined for their corresponding representations and relevance in Wikidata. The relevance between entities source and destination is calculated as:

$$\mathcal{R}_{ELF}^{(s,d)} = Relatedness(v_s^{wiki}, v_d^{wiki}) \tag{4}$$

where $v_s^{wiki}$ and $v_d^{wiki}$ are the entity representation of ($v_s^{amr}$ and $v_d^{amr}$) found in Wikidata. If $\mathcal{R}_{ELF}^{(s,d)}$ exceeds $\gamma$, the entities $v_s^{amr}$ and $v_d^{amr}$ are related in Wikidata. This implies the existence of potential evidence path to be attached between them.

**Context-Level Filtering (CLF)**: The CLF algorithm determines the relevance between $v_s^{amr}$ and $v_d^{amr}$. The CLF follows principles of $A^*$ search algorithm to search path in the knowledge graph from the starting entity $v_s^{amr}$ to the destination entity $v_d^{amr}$.

For each entity $v_i^{wiki} \in \mathcal{V}_{wiki}$ lying in the evidence path between $v_s^{amr}$ and $v_d^{amr}$, the relevancy is calculated between $v_i^{wiki}$ and $v_d^{amr}$ to predict the possibility of finding a rich evidence path between source and destination:

$$\mathcal{R}_{CLF}^{(s,d)} = Relatedness(v_i^{wiki}, v_d^{wiki}) = \frac{Tag(v_s^{wiki}, v_i^{wiki})}{n_i + \epsilon} + \frac{Tag(v_i^{wiki}, v_d^{wiki})}{n_m - n_i + \epsilon} \tag{5}$$

where $Tag(i,j)$ is the tagme score summed up from $i$ to $j$, $n_i$ and $n_m$ represent the current hopes and maximum hops respectively, a very small value, $\epsilon$ is added to avoid division by zero. The first term averages the Tagme score from source ($v_s^{wiki}$) to current node ($v_i^{wiki}$) and the second term averages the heuristic value of Tagme score from current node ($v_i^{wiki}$) to destination ($v_d^{wiki}$). If $\mathcal{R}_{CLF}^{(s,d)}$ is above the CLF threshold ($\delta$), the entity $v_i^{wiki}$ is linked with the next entity in the Wikidata path until $v_d^{amr}$ is reached. This process leads to the attachment of evidence information between entities, enriching the AMR graph with external knowledge.

This process is repeated for each pair of entities in $\mathcal{G}^{amr}$ and the resultant graph is called WikiAMR Graph. The detailed algorithm to generate WikiAMR using ELF and CLF is presented in Algo. 1.

---

**Algorithm 1** Evidence-Linking Algorithm

**Input:** AMR $\mathcal{G}^{amr}$, Wikigraph $\mathcal{G}^{wiki}$, $\gamma$, $\delta$
**Output:** WikiAMR graph $\mathcal{G}^{WikiAMR}$
  **function** ELF($\mathcal{G}^{amr}$, $\mathcal{G}^{wiki}$, $\gamma$, $\delta$):
    $evidence \leftarrow []$
    **for** each node pair $u, v \in \mathcal{G}^{amr}$ **do**
      Get relatedness $\mathcal{R}_{ELF}$ between
      start node $u$, and goal node $v$
      **if** ($\mathcal{R}_{ELF} > \gamma$): **then**
        $path = $ CLF($u, v, \gamma, \delta$);
        $evidence.append(path)$;
      **end if**
    **end for**
    Integrate $\mathcal{G}^{amr}$ and Relation paths to get $\mathcal{G}^{WikiAMR}$
    **return** $\mathcal{G}^{WikiAMR}$
  **end function**

**Algorithm 2** Context Level Filtering

  **function** CLF($start, goal, \gamma, \delta$):
    $path \leftarrow [start]$, $relation\_path \leftarrow []$
    **while** $path$ is $notempty$ **do**
      Pick $n_{cur}$, last indexed node from $path$
      **if** $n_{cur} = goal$ **then**
        Extract relationships from $\mathcal{G}^{wiki}$
        and append them in $relation\_path$
        **return** $relation\_path$
      **end if**
      **for** each neighbor $n_{adj}$ of $n_{cur}$ in $\mathcal{G}^{wiki}$
        **&** $n_{adj} \notin path$ **do**
        Get $\mathcal{R}_{CLF}$ between $n_{adj}, goal$
        **if** ($\mathcal{R}_{CLF} > \delta$) **then**
          Append $n_{adj}$ to $path$
        **end if**
      **end for**
    **end while**
  **end function**

---

**WikiAMR** is denoted as $\mathcal{G}^{WikiAMR}$, comprises interconnected undirected evidence paths between entities in $\mathcal{G}^{amr}$. This structure facilitates reasoning about the evidence present in external knowledge along with the directed acyclic structure extracted from the text document. Mathematically, it can be represented as follows:

$$\mathcal{G}^{WikiAMR} = \mathcal{G}^{amr} \cup \sum_{s,d} \mathcal{P}^{wiki}(v_s^{amr}, v_d^{amr}) \tag{6}$$

Here, $\mathcal{P}^{wiki}$ is the evidence path between $v_s^{amr}$ and $v_d^{amr}$ and $\sum$ denotes the graph generated from the set of evidence paths.

In continuation of above example, suppose we want to link the entities $Donald\ Trumph$ and $Mafia$, the relevance $\mathcal{R}_{ELF}$ between them is calculated using $Relatedness(\cdot)$ function. If $\mathcal{R}_{ELF} > \gamma$, the entities $Donald\ Trumph$ and $Mafia$ are identified as relevant, potentially holding evidence path between them. Upon identifying relevant entities, the $\mathcal{R}_{CLF}$ between the next entity $v_i^{wiki}$, $Media\ Coverage$ and the destination entity $Mafia$ is computed. If $\mathcal{R}_{CLF} > \delta$, entity $v_i^{wiki}$ is linked with entity $v_{i+1}^{wiki}$ in the AMR graph until the destination is reached. This results in the attachment of relevant evidence information. This is repeated over all the pair of entities in AMR and the final graph is represented as WikiAMR.

### 3.3.3 PATH-AWARE GRAPH LEARNING MODULE

This module plays a crucial role in EA$^2$N by generating informative features from the enriched WikiAMR graph obtained. These features capture essential semantic relationships and enable the model to gain a deeper understanding of the information present in the news articles. The module involves a Graph Transformer, which leverages multi-head attention mechanisms to process the WikiAMR representation in a way that facilitates effective reasoning and representation learning.

**Relation Path Encoder** The WikiAMR obtained from the above section is passed to the node intialization and relation encoder to get the encoding of WikiAMR in $\mathbb{R}^{N \times L \times D'}$ , where $D'$ is the dimension of the graph encoding.

To facilitate the model's recognition of explicit graph paths from $\mathcal{G}^{WikiAMR}$, the relation encoder is applied to capture the shortest path between two entities. The sequence representing this path is transformed into a relation vector using a Gated Recurrent Unit (GRU) based RNN (Cho et al., 2014). The mathematical representation of this relation encoding is given by:

$$\overrightarrow{p}_t = \text{GRU}_f(\overrightarrow{p}_{t-1}, sp_t) \quad \overleftarrow{p}_t = \text{GRU}_g(\overleftarrow{p}_{t+1}, sp_t)$$

Here, $sp_t$ signifies the shortest path of the relation between the two entities. As per Cai & Lam (2020)'s paper, in order to calculate the attention score, the final relational encoding $r_{ij}$ is divided into two separate encodings: $r_{i \to j}$ and $r_{j \to i}$, which are obtained using a linear layer and the parameter matrix $W_r$.

$$r_{ij} = [\overrightarrow{p}_n; \overleftarrow{p}_0], \quad [r_{i \to j}; r_{j \to i}] = W_r r_{ij}$$

The Graph Transformer processes the input $\mathcal{G}^{WikiAMR}$ using multi-head attention mechanism. Then the attention scores $\alpha_{ij}$ are computed based on both the entity representations and their relation representation.

$$\begin{aligned}
\alpha_{ij} = g(e_i, e_j, r_{ij}) &= (e_i + r_{i \to j})W_q^T W_k(e_j + r_{j \to i}) \\
&= \underbrace{e_i W_q^T W_k e_j}_{(a)} + \underbrace{e_i W_q^T W_k r_{j \to i}}_{(b)} + \underbrace{r_{i \to j} W_q^T W_k e_j}_{(c)} + \underbrace{r_{i \to j} W_q^T W_k r_{j \to i}}_{(d)}
\end{aligned} \tag{7}$$

The attention weights here are computed to work over entities based on relations and each term in Eq. 7 holds an explanation. The term (a) signifies content-based addressing, (b) and (c) capture the source-dependent and target-dependent relation bias, and (d) embodies a universal relation bias, encompassing a broader perspective on relation interactions. Collectively, this equation explains a comprehensive mechanism for the model to reason and weigh entity-relation interactions.

**Graph Transformer for Representation Learning**

Graph Transformer applies self-attention to capture dependencies between different positions within each WikiAMR graph representation. The encoder consists of multiple identical blocks, of which the core is multi-head attention. The model computes attention weights for the encoded paths to learn the enhanced representations. Given a set of attention heads H, each head computes distinct Query ($Q_i$), Key ($K_i$), and Value ($V_i$) matrices, which are then linearly combined through learnable weight matrices ($W_i$) to produce the final attended representation:

$$A_i = Attn(Q_i, K_i, V_i) = softmax\left(\frac{Q_i K_i^T}{\sqrt{D'}}\right) V_i \tag{8}$$

$$\mathrm{A} = Concat([\mathrm{A}_1, \mathrm{A}_2, ..., \mathrm{A}_h])W_{\mathrm{H}} \tag{9}$$

Here, $h$, $\mathrm{A}_i$ and $W_{\mathrm{H}}$ represent the number of attention heads, output of the $i^{th}$ head, and learnable weight matrix. This dynamic method enhance intricate semantic extraction.

After computing the attention weights, the Graph Transformer (GT) encodes the integrated Wiki-AMR representations $\mathcal{G}^{WikiAMR}$ as follows:

$$\mathcal{Z}^g = \mathrm{GT}(\mathcal{G}^{WikiAMR}, A) \in \mathbb{R}^{N \times L \times D'} \tag{10}$$

Where $\mathcal{Z}^g$ represents the final graph embedding obtained from the Graph Transformer, and $D'$ is the dimension of the feature vector.

## 3.4 CLASSIFICATION MODULE

The final stage of EA$^2$N involves the Classification Module, which takes the semantically-informed AMR representation and the enriched language features to produce the fake news predictions.

We concatenate the graph $\mathcal{Z}^g$ and language features $\mathcal{Z}^l$ to create the final fused embedding:

$$\mathcal{Z} = Concat([\mathcal{Z}^l, \mathcal{Z}^g]) \in \mathbb{R}^{N \times L \times (D+D')} \tag{11}$$

Finally, we pass $\mathcal{Z}$ through a classification transformer (CT) followed by a softmax layer to obtain the final probabilites $Y_{\mathrm{pred}}$ over real and fake.

$$f(Z) = softmax(\mathrm{CT}(\mathcal{Z})) \in \mathbb{R}^{N \times 2}, \ Y_{\mathrm{pred}} = argmax(f(Z)) \tag{12}$$

The comprehensive EA$^2$N model, leveraging AMR, external knowledge integration, affective features, and attention mechanisms, offers a powerful and novel approach to Fake News Detection.

## 4 EXPERIMENTAL SETUP

### 4.1 DATASET AND EVALUATION METRIC

In order to assess the effectiveness of EA$^2$N, we perform experiments on two benchmark datasets, namely, PolitiFact and GossipCop (Shu et al., 2020a). These datasets consist of 815 and 7,612 news articles, respectively, along with labels assigned by journalists and domain experts. Additional information regarding the preprocessing and implementation details can be found in Appendix A. We evaluate our model using a set of metrics, including Precision (Pre), Recall (Rec), F1-score, Accuracy (Acc), and Area Under the ROC curve (AUC). We conduct 5-fold cross-validation and report the average results.

### 4.2 BASELINES

In our evaluation, we contrast our EA$^2$N model with various state-of-the-art baselines, categorized into two groups. The first group utilizes only textual information (**SVM** (Yang et al., 2012), **DTC** (Castillo et al., 2011), **RFC** (Kwon et al., 2013), **GRU-2** (Ma et al., 2016), **FF** (FakeFlow) (Ghanem et al., 2021)), while the second incorporates auxiliary knowledge in addition to textual features (**B-TransE** (Pan et al., 2018), **KCNN** (Wang et al., 2018), **GCAN** (Lu & Li, 2020), **KAN** (Dun et al., 2021), **FinerFact** (Jin et al., 2022)).

Table 1: Comparative study of our model EA$^2$N w.r.t. different baselines.

| | Method | PolitiFact | | | | | GossipCop | | | | |
|---|---|---|---|---|---|---|---|---|---|---|---|
| | | Pre | Rec | F1 | Acc | AUC | Pre | Rec | F1 | Acc | AUC |
| T | SVM | 0.7460 | 0.6826 | 0.6466 | 0.6694 | 0.6826 | 0.7493 | 0.6254 | 0.5955 | 0.6643 | 0.6253 |
| | RFC | 0.7470 | 0.7361 | 0.7362 | 0.7406 | 0.8074 | 0.7015 | 0.6707 | 0.6691 | 0.6918 | 0.7389 |
| | DTC | 0.7476 | 0.7454 | 0.7450 | 0.7486 | 0.7454 | 0.6921 | 0.6922 | 0.6919 | 0.6959 | 0.6929 |
| | GRU-2 | 0.7083 | 0.7048 | 0.7041 | 0.7109 | 0.7896 | 0.7176 | 0.7079 | 0.7079 | 0.7180 | 0.7516 |
| | FF | 0.8462 | 0.7923 | 0.8193 | 0.8574 | 0.8627 | 0.7263 | 0.7352 | 0.7307 | 0.7563 | 0.7616 |
| T+K | B-TransE | 0.7739 | 0.7658 | 0.7641 | 0.7694 | 0.8340 | 0.7369 | 0.7330 | 0.7340 | 0.7394 | 0.7995 |
| | KCNN | 0.7852 | 0.7824 | 0.7804 | 0.7827 | 0.8488 | 0.7483 | 0.7422 | 0.7433 | 0.7491 | 0.8125 |
| | GCAN | 0.7945 | 0.8417 | 0.8345 | 0.8083 | 0.7992 | 0.7506 | 0.7574 | 0.7709 | 0.7439 | 0.8031 |
| | KAN | 0.8687 | 0.8499 | 0.8539 | 0.8586 | 0.9197 | 0.7764 | 0.7696 | 0.7713 | 0.7766 | 0.8435 |
| | FinerFact | 0.9196 | 0.9037 | 0.9172 | 0.9092 | 0.9384 | 0.8615 | 0.8779 | 0.8685 | 0.8320 | 0.8637 |
| Ours | EA$^2$N | **0.9333** | **0.9324** | **0.9328** | **0.9318** | **0.9523** | **0.8947** | **0.8865** | **0.8906** | **0.8713** | **0.9014** |

Table 2: Analysis on number of hops linked between entities.

| | # 1 hop | # 2 hops | # 3 hops | # 4 hops | # 5 hops |
|---|---|---|---|---|---|
| Politifact | 951 | 10 | 5 | 3 | 2 |
| Gossipcop | 5482 | 23 | 8 | 5 | 3 |

## 5 RESULTS

We used various transformer based model for textual encoding and reported the best results for EA$^2$N in the table. Table 1 shows a comparative analysis of EA$^2$N against various models. The standard deviations for accuracy and F1-score metrics in Politifact are 2.17 and 1.82, respectively and in Gossipcop, these standard deviations stand at 2.35 and 2.08. The table clearly demonstrates that our model, EA$^2$N, outperforms the state-of-the-art model, FinerFact, by 1.6%, 2.3% in terms of F1-score and accuracy on the Politifact, and by 2.2%, 3.9% on the Gossipcop, respectively. Interestingly, our model achieves these superior results without integrating social information, which FinerFact utilizes. Furthermore, our model surpasses KAN's performance on both datasets with F1-score and accuracy improvement of 7.9%, 8.1% and 11.9%, 9.5%. This is attributed to our model's ability to consider contextual information across multiple entities in the AMR and link evidence between them, unlike KAN, which only focuses on the contextual information of a single entity. This enables our model to learn the facts between entities, benefiting from external knowledge.

## 6 ABLATION STUDY

### 6.1 COMPARISON ON DIFFERENT LANGUAGE ENCODERS

We employ several transformer-based models to assess the effectiveness of EA$^2$N across various textual encodings. These include BERT-base (Devlin et al., 2019), RoBERTa-base (Liu et al., 2019), XLNET-base (Yang et al., 2019), and ELECTRA-base (Clark et al., 2020). The results of this evaluation, in conjunction with the baseline (FinerFact) result, are illustrated in Fig. 3. Notably, ELECTRA outperforms other models, exhibiting F1-score and accuracy of 0.9328, 0.9318 on the Politifact dataset and 0.8906, 0.8713 on the Gossipcop dataset. Comparing the remaining models, both XLNET and BERT demonstrate superior performance over RoBERTa. In comparison with baseline, this study concludes that by leveraging various textual encoders, EA$^2$N model surpasses other existing fake news detection models, yielding substantial improvements in performance.

### 6.2 COMPARISON ON DIFFERENT EA$^2$N VARIANTS

We conducted experiments with our model EA$^2$N (**LE|WikiAMR**) by incorporating different variations, including: 1) Language Encoder (**LE**) 2) AMR (**AMR**) and 3) Language Encoder with AMR (**LE|AMR**). Our findings from Fig. 4 indicate that only AMR model performs better than only LE model. Moreover, when we combine both language encoder and AMR (**LE|AMR**), there is a significant improvement of 6-8% observed over the only LE and AMR models. Additionally, when we integrate evidence in AMR into our final model (**LE|WikiAMR**), there is a further enhancement of 3-4% over the **LE|AMR** model for both datasets. We conducted two-tailed t-tests and observed a significant difference between EA$^2$N variants, obtaining a significance score(p-value) $< 0.01$ thus rejecting the null hypothesis. The detailed analysis of the same is covered in Appendix A.3.4.

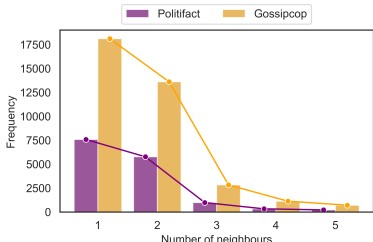

Figure 3: Comparison on different language encoders.

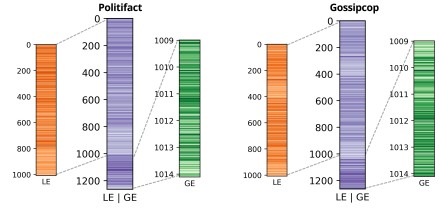

Figure 4: Comparison on different EA$^2$N variants.

Figure 5: Analysis on number of neighbours searched between entities.

Figure 6: Attention weight analysis over random samples.

### 6.3 ANALYSIS ON EVIDENCE LINKING ALGORITHM

In our proposed Evidence Linking Algorithm, we conducted an analysis of the number of hops linked and the number of neighbors searched between start and goal nodes. From Table 2, It is observed that the majority of entities where we found connections in both datasets are linked using 1 hop, indicating direct relations between them. As the number of hops increased, the proportion of linked entities gradually decreased, with very few entities linked using 5 hops relations. Also, in Fig. 5, the results revealed that most entities are searched within the first neighbor, and the frequency gradually decreases for subsequent neighbors. Based on findings, we capped linked path at 5 nodes and explored up to 5 neighbors for each node between start and goal entities in our algorithm.

### 6.4 ANALYSIS ON ATTENTION WEIGHTS

In order to investigate the influence of WikiAMR features on our model, we have conducted an examination of the attention weights from the final layer of the EA$^2$N. We delve into the attention weights of the Language Encoder (**LE**), the Graph Encoder (**GE**), as well as the combined Language Encoder with Graph Encoder (**LE|GE**). Analyzing the attention weights of **LE|GE** from Fig. 6, we deduce that the lower portion of the feature set (**GE** feature set) holds significant influence on the model's performance. This conclusion arises from the fact that our proposed WikiAMR encapsulates a comprehensive and intricate semantic structure of news articles. Furthermore, delving into the weights of individual encoders, we infer that within the **LE**, the initial feature subset strongly affects the model's behavior. This stems from the fact that the title of a news article, a concise summary of the news, is typically situated in the initial sentences. On the other hand, for **GE**, the entire feature set carries significance since WikiAMR emphasizes crucial semantic relationships among entities.

## 7 CONCLUSION

In this study, we introduce EA$^2$N, a novel Evidence-based AMR Attention Network designed to effectively identify fake news by harnessing external knowledge from Wikidata within the AMR graph through a proposed evidence linking algorithm. For the future direction, we are interested to explore more ways to encode external knowledge using social information with semantic relations in AMR. It is important to note that our study not only provide solution for fake news detection but it has the potential to pave the way for solving various other NLP applications.

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

## A  APPENDIX

### A.1  EXPERIMENTAL SETUP

#### A.1.1  DATASET

Details of the datasets are provided in Table 3. Basic prepossessing steps such as removal of '@', '#' symbols, removal of website URLs etc., are performed on the news content.

Table 3: Distribution of data for a) Politifact, b) Gossipcop.

|            | # True | # Fake | # Total |
|------------|--------|--------|---------|
| PolitiFact | 443    | 372    | 815     |
| GossipCop  | 4219   | 3393   | 7612    |

#### A.1.2  IMPLEMENTATION DETAILS

To perform Entity and Context Level Filtering (ELF and CLF) for linking external knowledge, we utilize the $TagMe$ API (Ferragina & Scaiella, 2010). In order to generate the AMR graph, we have used pretrained STOG model (Zhang et al., 2019). For filtering at the entity and context levels, we set the values of $\gamma$ and $\delta$ to 0.4 and 0.1, respectively. These values assume that primary entities in the AMR should exhibit high relatedness, while their neighbors will have comparatively lower relatedness. During the evidence linking process, we limit the linked path to a maximum of 5 hops $(n_m)$ including start and goal nodes and explored up to 5 neighbors for each node between the primary entities. This limitation is kept to enforce time and memory constraint on our algorithm because of large number of entities present in the dataset. Further, the ablation study supports our choice of 5 for both the cases. We employ transformer encoding with an embedding size of 512. Additionally, the affective lexical features are set to an embedding size of 240 using 10 segments in a news article. To facilitate the learning rate annealing, we adopt the cosine learning rate technique (Loshchilov & Hutter, 2017). This approach allows the learning rate to anneal from an initial learning rate to a maximum learning rate and then further down to a minimum learning rate, which is substantially lower than the initial learning rate. The parameters used for the graph path learning model are identical to those described in Cai & Lam (2020)'s model. The training of our model was conducted on an NVIDIA A30 GPU with 24 GB of memory.

## A.2 SUPPLEMENTARY EXPERIMENTS

### A.2.1 EFFECTS OF DIFFERENT LEARNING RATES ON EA$^2$N

In this module, we explore the impact of Learning Rate (LR) variations on the performance of our EA$^2$N model. We investigate three LR scheduling strategies: **Linear**, **OneCycleLR** (Smith & Topin, 2018), and **Cosine** (Loshchilov & Hutter, 2017). These strategies are systematically evaluated on both the Politifact and Gossipcop datasets to discern their influence on the model's accuracy and F1 score.

Table 4: Analysis on different learning rate strategies.

| Learning rate | Poltifact | | Gossipcop | |
|---|---|---|---|---|
| | F1 | Acc | F1 | Acc |
| Linear | 0.8869 | 0.8748 | 0.8571 | 0.8549 |
| OneCycleLR | 0.9112 | 0.9087 | 0.8726 | 0.8636 |
| Cosine | 0.9328 | 0.9318 | 0.8906 | 0.8713 |

The experimented results from Table 4 unveils notable trends in performance across different LR schedules. The **Cosine LR** stands out as the most superior strategy on both datasets. For instance, on the Politifact dataset, **Cosine LR** achieves F1 scores and accuracies of 0.9328 and 0.9318, respectively. In a similar vein, on the Gossipcop dataset, **Cosine LR** boasts an F1 score of 0.8906 and an accuracy of 0.8713. It's worth noting that **OneCycleLR** follows closely, demonstrating an impressive performance catch-up, eventually surpassing Linear LR. For example, **OneCycleLR** achieves F1 scores of 0.9112 and 0.8726 on the Politifact and Gossipcop datasets, respectively, outpacing **Linear LR**'s corresponding scores of 0.8869 and 0.8571.

However, it's important to highlight a nuanced aspect. Despite the superior performance of **Cosine LR**, the **OneCycleLR** strategy exhibits the ability to reach saturation levels more swiftly in fewer epochs due to its dynamic LR scheduling. This introduces a delicate trade-off between the time efficiency of **OneCycleLR** and the enhanced performance of **Cosine LR**.

### A.2.2 EFFECTS OF DIFFERENT BATCH SIZES ON EA$^2$N

This study delves into the effects of varying batch sizes on the performance of our EA$^2$N model. Batch sizes of 1, 2, 4, and 8 are systematically tested on both the Politifact and Gossipcop datasets to gauge their impact on accuracy and F1 score. Table 5 underscores the relationship between batch

Table 5: Analysis on different batch sizes.

| Batch size | Poltifact | | Gossipcop | |
|---|---|---|---|---|
| | F1 | Acc | F1 | Acc |
| 1 | 0.8807 | 0.8591 | 0.8532 | 0.8471 |
| 2 | 0.8974 | 0.8711 | 0.8609 | 0.8581 |
| 4 | 0.9178 | 0.9228 | 0.8743 | 0.8685 |
| 8 | 0.9328 | 0.9318 | 0.8906 | 0.8713 |

size and model performance. Generally, larger batch sizes exhibit improved results. For example, on the Politifact dataset, the F1 scores and accuracies increase as batch size grows: from 0.8807 and 0.8591 for a batch size of 1 to 0.9328 and 0.9318 for a batch size of 8. The same trend holds for the Gossipcop dataset: F1 scores and accuracies rise from 0.8532 and 0.8471 for batch size 1 to 0.8906 and 0.8713 for batch size 8. However, intriguingly, beyond a certain threshold, the performance gains start to diminish. For instance, on Politifact, as the batch size increases from 4 to 8, the F1 score only marginally improves from 0.9178 to 0.9328, highlighting the intricate interplay between batch size and model training dynamics.

## A.3 ERROR CASE ANALYSIS

**Case Study:** *Executive Order Leads to Capture of ISIS Leader Rasheed Muhammad Search tags for this page ag apologizes rasheed muhammad .... terror suspect rasheed .... was captured.*

The sentence in focus is: "Executive Order Leads to Capture of ISIS Leader Rasheed Muhammad..." Corresponding to this, an Abstract Meaning Representation (AMR) graph is constructed to represent the semantic structure. However, this particular AMR graph exhibits some limitations in its representation.

$$
\begin{aligned}
&(snt1\ (l\ /\ \text{lead-03} \\
&\qquad :ARG0\ (o\ /\ \text{order-01} \\
&\qquad\qquad :mod\ (e\ /\ \text{executive})) \\
&\qquad :ARG2\ (c\ /\ \text{suspect-01} \\
&\qquad\qquad :ARG1\ (p\ /\ \text{person} \\
&\qquad\qquad\qquad :name\ (n\ /\ \text{name} \\
&\qquad\qquad\qquad\qquad :op1\ \text{``Rasheed''} \\
&\qquad\qquad\qquad\qquad :op2\ \text{``Muhammad''})) \\
&\qquad\qquad\qquad :ARG0-of\ (l2\ /\ \text{lead-02} \\
&\qquad\qquad\qquad\qquad :ARG1\ (c2\ /\ \text{criminal-organization} \\
&\qquad\qquad\qquad\qquad\qquad :name\ (n2\ /\ \text{name} \\
&\qquad\qquad\qquad\qquad\qquad\qquad :op1\ \text{``Islamic''} \\
&\qquad\qquad\qquad\qquad\qquad\qquad :op2\ \text{``State''}))))))) \\
&\quad\ldots \\
&\quad\ldots \\
&\quad\ldots \\
&\qquad :op5\ (p9\ /\ \text{person} \\
&\qquad\qquad :name\ (n9\ /\ \text{name} \\
&\qquad\qquad\qquad :op1\ \text{``Rasheed''} \\
&\qquad\qquad\qquad :op2\ \text{``Muhammad''})) \\
&\qquad\qquad :ARG2-of\ (s2\ /\ \text{suspect-01} \\
&\qquad\qquad\qquad :ARG1\ (t4\ /\ \text{terror}))))
\end{aligned}
$$

### A.3.1 LIMITATION IN AMR GRAPH CONSTRUCTION

The limitation arises from an incomplete AMR graph construction. Specifically, towards the end of the graph, only the 'suspect' part is reflected, omitting the 'captured' aspect. This incompleteness affects the salient features of the graph and consequently impacts the accuracy of our model. It hinders a holistic understanding of the event, overlooking crucial actions like 'capture', which are vital for accurate predictions.

### A.3.2 INCONSISTENT RELEVANCE SCORES IN $TagMe$

Furthermore, another aspect of concern is the inconsistent relevance scores in $TagMe$, an entity linking tool. These scores play a vital role in the Entity-Level Filtering (ELF) and Context-Level Filtering (CLF) stages of our model. When the relevance scores are inconsistent, they affect the path determination process in both the stages. For instance, relevance scores for certain prominent entities are not considered if they are below the thresholds, which were initially tuned for them. This alteration in the path can significantly impact the accuracy and reliability of the model's predictions.

### A.3.3 STUDY ON SNOPES DATASET

In order to evaluate the generalizability of our model, we additionally conducted tests on the publicly available Snopes dataset Vo & Lee (2020), consisting of a total of 1703 fact-checking articles covering various political topics sourced from the fact-checking website *snopes.com*. This dataset encompasses multiple classes, including false, true, mostly false, mostly true, scam, unknown, etc. Given that EA$^2$N is a fake/real classification model, we focused exclusively on the true and false classes. For the final assessment, we utilized 1106 articles classified as fake and 182 articles classified as true. To address class imbalance in the dataset, we employed standard NLP based data augmentation techniques. The results presented in Table 6 clearly demonstrate that our model EA$^2$N surpasses other models by a significant margin. We have compared our model EA$^2$N (**LE|WikiAMR**) with 1) FakeFlow (**FF**) 2) Language Encoder (**LE**) 3) AMR (**AMR**) 4) Language Encoder with AMR (**LE|AMR**). It is evident form the results that when we integrate evidence in AMR into our final

Table 6: Comparative study on Snopes dataset.

| Method | F1 | Acc |
|---|---|---|
| FF | 0.8712 | 0.8542 |
| LE | 0.8620 | 0.8511 |
| AMR | 0.8961 | 0.8769 |
| LE\|AMR | 0.9144 | 0.8869 |
| LE\|WikiAMR (EA$^2$N) | **0.9212** | **0.9045** |

Table 7: T-test for EA$^2$N variants in Politifact and Gossipcop datasets.

| Method | Politifact | | Gossipcop | |
|---|---|---|---|---|
| | t-statistics | p-value | t-statistics | p-value |
| LE and AMR | 6.19 | 3.39e-09 | 8.21 | 2.64e-14 |
| AMR and LE\|AMR | 14.39 | 1.30e-32 | 14.46 | 7.96e-33 |
| LE\|AMR and LE\|WikiAMR | 12.40 | 1.63e-26 | 6.85 | 8.74e-11 |

model (**LE|WikiAMR**), there is 2-5%, 1-6% gain in accuracy and F1- score respectively from all the other models.

### A.3.4 TWO TAILED T-TESTS ON EA$^2$N VARIANTS

We conducted two-tailed t-tests to evaluate the significance of differences between the accuracy of EA$^2$N variants on randomly selected samples. The hypotheses were defined as follows:

- **Null Hypothesis (H$_0$):** There is no significant difference between the accuracy of EA$^2$N variants on randomly selected samples.
- **Alternative Hypothesis (H$_1$):** There is a significant difference between the accuracy of EA$^2$N variants on randomly selected samples.

The t-statistic was calculated using the formula:

$$t = \frac{\bar{x1} - \bar{x2}}{\sqrt{\frac{s1^2}{n1} + \frac{s2^2}{n2}}}$$

Here $s1$ and $s2$ are the standard deviation, $n1$ and $n2$ represent the samples considered, and $\bar{x1}$ and $\bar{x2}$ represent the mean accuracy for Model 1 and Model 2, respectively.

The t-statistic scores and p-values for all the models based on randomly selected 100 sample sets, grouped by dataset, are presented in Table 7. Standard deviation for individual models such as LE, AMR, LE|AMR, LE|WikiAMR are 2.53, 3.21, 2.32, 2.71 for Politifact and 2.11, 3.07, 2.73, 2.35 for Gossipcop, respectively. It is evident from the table that the obtained significance values for the corresponding variants are less than 0.01, contradicting the null hypothesis. This statistical interpretation indicates a significant difference between the evaluated variants in both the Politifact and Gossipcop datasets.

### A.3.5 SENSITIVITY ANALYSIS ON $\gamma$ AND $\delta$

We performed a sensitivity analysis to examine the influence of different ELF (Entity Level Filtering) and CLF (Context Level Filtering) thresholds on model's performance. The average entity values for ELF thresholds ($\gamma$) and CLF thresholds ($\delta$), based on 100 samples, are displayed in Fig. 7.

The threshold values significantly impact system behavior. Higher thresholds lead to fewer entities, indicating stricter linking and classification criteria. When entities decrease, the graph tends to converge from WikiAMR to AMR, potentially reducing accuracy. Conversely, a higher entity count results in a larger graph size, escalating training times. Optimizing threshold values involves striking a balance between accuracy and computational efficiency. The ELF threshold at 0.4 and the CLF threshold at 0.1 emerge as potential optimal values. A $\gamma$ of 0.4 maintains a balance between accurate and manageable linking of primary entities. Meanwhile, a $\delta$ of 0.1 helps control relevancy of the linked paths without compromising accuracy significantly. Our threshold selection aims to strike

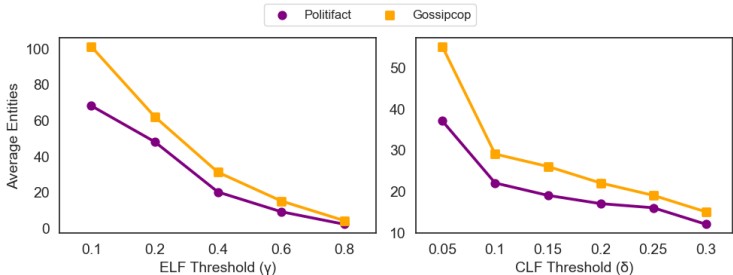

Figure 7: Sensitivity analysis on $\gamma$ and $\delta$.

a balance between graph size, linking time, training time, and accuracy. It's essential to avoid excessively stringent thresholds, which could diminish accuracy, and excessively lax thresholds, which might inflate computational costs. Additionally, high entity counts don't guarantee increased accuracy; they may contain redundancies. Further experiments are necessary to fine-tune thresholds and assess their influence on overall model performance.

