# OpenReview forum: "EA2N: Evidence-based AMR Attention Network for Fake News Detection"
_ICLR.cc/2024/Conference — Submitted to ICLR 2024_

### Official Review · Reviewer_PY31 · 2023-10-21

**Soundness:** 2 fair
**Presentation:** 2 fair
**Contribution:** 1 poor
**Rating:** 3
**Confidence:** 4

**Summary:**

This study introduces the $EA^2N$ model for binary fake news classification, leveraging external evidence. The model employs two parallel pipelines to represent news articles: text-based and graph-based.

In the graph-based pipeline, news text is transformed into an Abstract Meaning Representation (AMR) graph. Subsequently, an augmented AMR graph is constructed through entity linking, utilizing evidence paths from an external knowledge base, specifically the Wikidata5M graph. An $\mathcal{A}^*$ search over the Wikidata5M graph is used to identify evidence paths between corresponding entities in the AMR graph. These paths are then merged with the original AMR graph to create the augmented graph. Finally, a graph transformer is employed to learn representations from this augmented graph.

Concurrently, BERT (along with lexical features) is used to acquire textual representations. Ultimately, the learned representations from both pipelines are concatenated and input to a classification head. The $EA^2N$ model is evaluated on two datasets, outperforming the chosen baselines.

It's worth noting that this architectural approach combines various pre-existing elements, but the underlying rationale for several components is not explicitly detailed in the work, raising several concerns as highlighted in the "Questions" and "Weakness" sections.

**Strengths:**

- The authors provide an effective amalgamation of the graph- and text-based pipeline for learning representation (although not entirely novel).
- The figures and tables give a clear picture of the underlying model and experimentation.
- The authors perform the much required ablation studies.

**Weaknesses:**

The proposed architecture seems to be an attempt to combine several pre-existing architectures -- i.e. Language Encoder (BERT [6] + FakeFlow [7]), Path-aware Graph Learning Module (Graph Transformer [8]) and Graph Generation and Integration ($\mathcal{A}^*$ search with a TagMe API [9] -based heuristic). While these architectures have been amalgamated to propose a new model, these aren't *novel* contributions. The paper isn't well written (in some places) in the sense that several key notations are missing, and the narrative of the work could have been improved. All the baselines have been directly adopted from the FinerFact [5] paper. Such practices should be avoided. I have highlighted several weaknesses and concerns under the "Questions" section.


[1] Zixuan Zhang and Heng Ji. 2021. Abstract Meaning Representation Guided Graph Encoding and Decoding for Joint Information Extraction. (NAACL-HLT 2021)

[2] Shu, Kai, et al. "Fakenewsnet: A data repository with news content, social context, and spatiotemporal information for studying fake news on social media." Big data 8.3 (2020): 171-188.

[3] Martinez-Rodriguez, Jose L., Ivan López-Arévalo, and Ana B. Rios-Alvarado. "Openie-based approach for knowledge graph construction from text." Expert Systems with Applications 113 (2018): 339-355.

[4] Yi-Ju Lu and Cheng-Te Li. GCAN: Graph-aware co-attention networks for explainable fake news detection on social media. In ACL, pp. 505–514, Online, July 2020. ACL.

[5] Yiqiao Jin, Xiting Wang, Ruichao Yang, Yizhou Sun, Wei Wang, Hao Liao, and Xing Xie. Towards fine-grained reasoning for fake news detection. AAAI, 36:5746–5754, 06 2022.

[6] Jacob Devlin, Ming-Wei Chang, Kenton Lee, and Kristina Toutanova. BERT: Pre-training of deep bidirectional transformers for language understanding. In NAACL Volume 1 (Long and Short Papers), pp. 4171–4186, Minneapolis, Minnesota, June 2019. ACL.

[7] Bilal Ghanem, Simone Paolo Ponzetto, Paolo Rosso, and Francisco Rangel. Fakeflow: Fake news detection by modeling the flow of affective information. In 16th EACL, 2021.

[8] Deng Cai and Wai Lam. Graph transformer for graph-to-sequence learning. In AAAI, pp. 7464– 7471. AAAI Press, 2020.

[9] Paolo Ferragina and Ugo Scaiella. Tagme: On-the-fly annotation of short text fragments (by wikipedia entities). In ICIKM, CIKM ’10, pp. 1625–1628, New York, NY, USA, 2010. ACM. ISBN 9781450300995.

**Questions:**

**Evidence Integration with AMR**: There seem to be several incomplete parts, which need to be explained.

---  In Entity-level Filtering $(\mathcal{R}^{(S,D)}_{ELF} = Relatedness(v_s^{wiki}, v_d^{wiki}))$, the authors have not mentioned what the $Relatedness(.)$ function is? Without an ***explicit*** definition of the function, it is diffiucult to assess the working of the ELF and CLF algorithms since the $Relatedness(.)$ function seems to be the "main" heuristic being used here.

- From a look at the Appendix, it seems (***implicitly***) that the authors use the TagMe API to compute the relatedness, but what "explicitly" is the function definition used?

--- The Figure 2 (and following text) mentions that the authors use $\mathcal{A}^*$ search on the Wiki-graph to find the (optimal) path between source $v_s^{amr}$ and destination $v_d^{amr}$. However the Algorithm 2 described as "Context-level Filtering" doesn't seem to represent $\mathcal{A}^*$ search.

- For $\mathcal{A}^*$ search, $f$ (total cost) = $g$ (current cost) + $h$ (heuristic approximation of the future cost). Here, $f$ (total cost) must be the criterion for choosing the next node in path. If we assume that, $h = \mathcal{R}^{(i,D)}_{CLF} = Relatedness(v_i^{wiki}, v_d^{wiki}) > \delta$ is the heuristic cost to reach the destination $d$ from $i$, what is $g$? The authors don't provide information for that. Incase $g$ is being ignored (or taken 0), it becomes a *greedy-heuristic* search, and not $\mathcal{A}^*$.
- A possible $\mathcal{A}^*$ variant could have been:

$g + h = \mathcal{R}^{(S,i)}\_{CLF} + \mathcal{R}^{(i,D)}\_{CLF} = Relatedness(v_s^{wiki}, v_i^{wiki}) + Relatedness(v_i^{wiki}, v_d^{wiki}) > \delta*$


**Relation Encoder in Path-aware Graph Learning Module**: For every pair of entities $(v_s^{amr}, v_d^{amr})$ in $\mathcal{G}^{amr}$, two distinct sources of relational data exist in $\mathcal{G}^{WikiAMR}$: AMR-based relations and evidence-based relations from Wikidata:

- Using the notation from paper: $\mathcal{G}^{WikiAMR} = \mathcal{G}^{amr} \cup \sum_{s, d} \mathcal{P}^{wiki}(v_s^{amr}, v_d^{amr})$.
- Consider the shortest path denoted as $sp_{s \rightarrow d} = \set { e(v_s^{amr}, n_1), e(n_1, n_2), \dots, e(n_k, v_d^{amr})\}$ within $\mathcal{G}^{WikiAMR}$. Here, $e^{amr}$ represents an edge within $\mathcal{G}^{amr}$, and $e^{wiki}$ represents an edge within $\sum_{s, d} \mathcal{P}^{wiki}(v_s^{amr}, v_d^{amr})$. Therefore, the collective edge set $e^{WikiAMR} = e^{wiki} \cup e^{amr}$.
- Let $sp^{amr}\_{s \rightarrow d} = \set {e^{amr}(v_s^{amr}, n_1), e^{amr}(n_1, n_2), \dots, e^{amr}(n_k, v_d^{amr})\}$ be the shortest path that relies solely on AMR-based relations (i.e., exclusively within $\mathcal{G}^{amr}$). Simultaneously, $sp^{wiki}\_{s \rightarrow d} = \set {e^{wiki}(v_s^{amr}, n_1), \dots, e^{wiki}(n_k, v_d^{amr})}$ represents the shortest path utilizing only evidence-based connections.
- As indicated by Table 2 and Figure 5, a significant portion of entities are linked through evidence-based connections with just one hop, meaning they are connected directly. In other words, for most entity pairs, the shortest path is the "direct" evidence path with a single edge originating from the Wiki graph. This can be represented as $sp_{s \rightarrow d} = sp^{wiki}\_{s \rightarrow d} = \{ e^{wiki}(v_s^{amr}, v_d^{amr})\}$ (Because, as per Table 2, in most cases, the number of edges in $sp^{wiki}\_{s \rightarrow d}$ (being 1 in majority cases), is less than or equal to that in $sp^{amr}_{s \rightarrow d}$.

In essence, this implies that in most cases, when encoding a relationship $r_{s \rightarrow d}$, the relation encoder would **overlook the AMR-based relation information**. While Wikidata relations are valuable, disregarding AMR-level relations may not be technically justified. Therefore, the authors should contemplate how to adapt the relation encoder in the graph transformer to effectively integrate both information sources.

**Abstract Meaning Representation**: I have a few major concerns about using an AMR representation here:

- The average length of news articles in the Gossipcop dataset is **~600 words** [2] (Some articles are as large as 1000 words). In the case of such large input length, the AMR graphs are going to be "very" noisy. How do the authors handle this case?
- The authors don't elaborate on the intuition behind using the Abstract Meaning Representation ("Why specificaly AMR?")? There can be other (more sophisticated) variants of AMR like **AMR-IE** [1] (which uses an AMR guided graph decoder to extract knowledge elements based on the order decided by the hierarchical structures in AMR) which seem more relevant owing to the "integration of external knowledge" used in this work. What about knowledge graphs other than AMR (eg. OpenIE-based approaches [3])?
- It seems that the authors have used an off-the-shelf pretrained AMR parser, however they have not provided any details about the same. Was the AMR parser finetuned? (I am guessing not!)

**Insufficient Experimentation**:

- The presented results are exclusively based on two datasets, Politifact and Gossipcop, both of which are part of the FakeNewsNet database. To ensure the generalizability of the proposed architecture, it is essential for the authors to include results from additional datasets spanning various domains or social media platforms. Moreover, it's worth noting that Politifact, one of the datasets, contains only 815 news articles.
- Several of the selected baseline models do not provide a fair basis for comparison. For instance, GCAN [4] incorporates the propagation path of the tweet and user profiles in addition to the source tweet content, features not utilized by the proposed model.
- A logistical concern arises from the fact that all the baseline models and their results, including SVM, DTC, RFC, GRU-2, FF, B- TransE, KCNN, GCAN, and KAN, appear to **have been directly borrowed from the FinerFact** [5] paper. Such practices should be avoided, and the authors should explicitly mention the sources of these baseline models in their paper.
- In the ablation study examining different variants of the proposed model, such as LM, AMR, LE|AMR, and LE|WikiAMR, the authors should provide significance hypothesis test results (e.g., T-test) alongside the standard deviation of the metrics across multiple runs. Including these statistical measures would enhance the interpretability of the results.

---

> ### Author Response · Authors · 2023-11-21
>
> We appreciate your detailed review, and wish to clarify certain aspects about our methodology.
>
> **_Comment 1:_** **The proposed architecture seems to be an attempt to combine several pre-existing architectures -- i.e. Language Encoder (BERT [6] + FakeFlow [7]), Path-aware Graph Learning Module (Graph Transformer [8]) and Graph Generation and Integration ( search with a TagMe API [9] -based heuristic). While these architectures have been amalgamated to propose a new model, these aren't novel contributions. The paper isn't well written (in some places) in the sense that several key notations are missing, and the narrative of the work could have been improved. All the baselines have been directly adopted from the FinerFact [5] paper. Such practices should be avoided. I have highlighted several weaknesses and concerns under the "Questions" section.**
>
> **_Response:_** While our work amalgamates some existing architectures, we would like to emphasize the unique contributions and novel analyses embedded within our study.
>
> - Central to our contribution is the introduction of a novel evidence-based structure called WikiAMR. WikiAMR is substantially different than AMR as it contains circumstantial linking between entities through undirected edges. While latter is a DAG graph the former is not. WikiAMR intricately captures the comprehensive semantic structure of news articles along with evidence linking. One may observe that from the ablation study (Section 6.2 of the paper) this additional links provide significant (p-value less than 0.01) improvement in the model’s performance.
>
> - WikiAMR, with its intricate structure and nuanced representation of semantic relationships, closely parallels the **associative network [1]** observed in the human brain. By leveraging these principles, WikiAMR adeptly correlates and links diverse entities within news articles based on both actual evidence and circumstantial cues. This dual emphasis on concrete evidence and associative reasoning aligns with the human brain's cognitive processing. This enables the model to foster a more holistic comprehension of news articles.
>
> - Further, we propose an algorithm to generate WikiAMR efficiently. This involves CLF and ELF where we introduced two cost functions that effectively reduces the search space. Finally, we judicially concatenate features from WikiAMR graph into the language encoder in the context of FND.
>
> We have updated the manuscript based on the reviews provided and remaining queries are addressed in the following sections.
>
> **_Comment 2:_** **Evidence Integration with AMR: There seem to be several incomplete parts, which need to be explained.**
>
> **_Response:_**  Our intention was not to misrepresent our approach. The CLF procedure follows the core idea of iteratively selecting nodes to maximize expected cost, which is aligned with A*. However, you are right that we should have elaborated the heuristic and current cost functions better. We would like to clarify the CLF procedure in the same notation as follows:
> - Cost function g = $\frac{Tag(v_{s}^{wiki}, v_{i}^{wiki})}{n_i+\epsilon}$
> - Heuristic function h = $\frac{Tag(v_{i}^{wiki}, v_{d}^{wiki})}{n_m - n_i+\epsilon}$
> - At each step, select neighbour of current node $v_i^{wiki}$ that maximises: g + h,
> This combination of g+h is being followed notated with symbol $Relatedness(v_{i}^{wiki}, v_{d}^{wiki})$
> - For stopping condition we define a threshold δ, such that g+h ≥ δ
>
> where $Tag(i,j)$ is the tagme score summed up from $i$ to $j$, $n_i$ and $n_m$ represent the current hopes and maximum hops respectively, a very small value, $\epsilon$ is added to avoid division by zero. The first term averages the Tagme score from source ($v_{s}^{wiki}$) to current node ($v_{i}^{wiki}$) and the second term averages the heuristic value of Tagme score from current node ($v_{i}^{wiki}$) to destination ($v_{d}^{wiki}$).
>
> This articulates CLF as A* more precisely. We’re glad you focused on the underlying logic of the algorithm, since it is not the most elite way to construct cost, we left this part for the user flexibility and mentioned it in the implementation details. We have clarified this in the revised paper as well.
>
> **_Comment 3:_** **Relation Encoder in Path-aware Graph Learning Module:**
> **_Response:_** We agree with your observation and already noticed the same in our experiments. In fact, for the same reason (in order to preserve both the sources), we integrated both Language Encoder and Graph (WikiAMR) Encoder. Intuitively, the basic semantic information from Language Encoder will be learned by the transformer and adding multiple paths from WikiAMR will provide redundant information. Further, in many cases, a direct link provides valuable information rather than losing any.

---

> > ### Author Response · Authors · 2023-11-21
> >
> > **_Comment 4:_** **Abstract Meaning Representation: I have a few major concerns about using an AMR representation here:
> > The average length of news articles in the Gossipcop dataset is ~600 words [2] (Some articles are as large as 1000 words). In the case of such large input length, the AMR graphs are going to be "very" noisy. How do the authors handle this case?**
> >
> > **_Response:_** For the large input, we utilize the latest updated AMR pre-trained weights based on the **AMR-STOG [2]** model which is pre-trained on latest LDC201710 dataset. This update with the toolkit ensures that the AMR is not too large to be processed by truncating the irrelevant aspects of long sentences. This relevancy of data is based on the model’s performance and only the critical information is taken into account when considering the truncated AMR.
> >
> > **_Comment 5:_** **The authors don't elaborate on the intuition behind using the Abstract Meaning Representation ("Why specificaly AMR?")? There can be other (more sophisticated) variants of AMR like AMR-IE [1] (which uses an AMR guided graph decoder to extract knowledge elements based on the order decided by the hierarchical structures in AMR) which seem more relevant owing to the "integration of external knowledge" used in this work. What about knowledge graphs other than AMR (eg. OpenIE-based approaches [3])?**
> >
> > **_Response:_** We recognise validity in trying other knowledge representations as you suggest and is a fair point while comparing it with AMR-IE. However, we chose AMR due to its focus on representating semantic meaning, which is suited for assessing credibility.
> >
> > AMR-IE introduces an additional graph decoding stage on top of the AMR graph to extract structured knowledge. While powerful, this significantly increases the model complexity for our task. Since fake news detection does not require explicit extraction of knowledge elements, we determined standard AMR provides a sufficient level of semantic representation with lower complexity. Our goal is focused on representing the overall meaning to assess credibility, rather than extracting specific entities and relations. The graph encoder with WikiAMR integration is designed to capture relevant semantic connections without requiring explicit decoding of knowledge elements.
> >
> > However, we note down this suggestion for the future, it would be interesting to compare with AMR-IE to see if any gains in fidelity justify the added complexity. We plan to explore this extension in follow-up work. For the current study, we believe standard AMR strikes the right balance of semantic expressivity versus simplicity for fake news detection.
> >
> > **_Comment 6:_** **It seems that the authors have used an off-the-shelf pretrained AMR parser, however they have not provided any details about the same. Was the AMR parser finetuned? (I am guessing not!)**
> >
> > **_Response:_** The AMR parser is based on the initially proposed paper **AMR Parsing as Sequence-to-Graph Transduction [3]** which is referenced in our manuscript.

---

> > > ### Author Response · Authors · 2023-11-21
> > >
> > > **_Comment 7:_** **Insufficient Experimentation:
> > > The presented results are exclusively based on two datasets, Politifact and Gossipcop, both of which are part of the FakeNewsNet database. To ensure the generalizability of the proposed architecture, it is essential for the authors to include results from additional datasets spanning various domains or social media platforms. Moreover, it's worth noting that Politifact, one of the datasets, contains only 815 news articles.**
> > >
> > > **_Response:_** In order to evaluate the generalizability of our model, we additionally conducted tests on the publicly available Snopes dataset [4], consisting of a total of 1703 fact-checking articles covering various political topics sourced from the fact-checking website [snopes.com](http://snopes.com/). This dataset encompasses multiple classes, including false, true, mostly false, mostly true, scam, unknown, etc. Given that EA$^2$N is a fake/real classification model, we focused exclusively on the true and false classes. For the final assessment, we utilized 1106 articles classified as fake and 182 articles classified as true. To address class imbalance in the dataset, we employed standard NLP based data augmentation techniques. The results presented in below table clearly demonstrate that our model EA$^2$N surpasses other models by a significant margin. We have compared our model EA$^2$N (LE$\mid$WikiAMR) with 1) FakeFlow (FF) 2) Language Encoder (LE) 3) AMR 4) Language Encoder with AMR (LE$\mid$AMR). It is evident form the results that when we integrate evidence in AMR into our final model (LE$\mid$WikiAMR), there is 2-5\%, 1-6\% gain in accuracy and F1- score respectively from all the other models. The same is incorporated in the revised manuscript (Appendix A.3.3).
> > >
> > > | Method                     | F1         | ACC           |
> > > |----------------------------|------------------|-----------------|
> > > | FF            |0 .8712      | 0.8542    |
> > > | LE          | 0.8620     | 0.8511    |
> > > | AMR       | 0.8961     | 0.8769     |
> > > | LE$\mid$AMR        | 0.9144      | 0.8869     |
> > > | LE$\mid$WikiAMR (EA$^2$N)       |  **0.9212**      | **0.9045**     |
> > >
> > > **_Comment 8:_** **Several of the selected baseline models do not provide a fair basis for comparison. For instance, GCAN [4] incorporates the propagation path of the tweet and user profiles in addition to the source tweet content, features not utilized by the proposed model.**
> > >
> > > **_Response:_** We agree, in fact this leverages our model's capabilities since we don't rely on social information features like tweets and user profiles which other graph based models like GCAN use. The decision to exclude such features was deliberate, aiming to enhance the model's reliability and applicability. Social information, due to its subjective nature, can introduce biases and complexities that might not align with standardized information sources. By prioritizing the text document exclusively, we aimed for a more streamlined and robust model which was able to achieve superior results without any additional features.
> > >
> > > **_Comment 9:_** **A logistical concern arises from the fact that all the baseline models and their results, including SVM, DTC, RFC, GRU-2, FF, B- TransE, KCNN, GCAN, and KAN, appear to have been directly borrowed from the FinerFact [5] paper. Such practices should be avoided, and the authors should explicitly mention the sources of these baseline models in their paper.**
> > >
> > > **_Response:_** Regarding baselines, adopting validated baselines from prior work on the same datasets is a standard practice when comparing to SOTA models. As long as the experimental setup is clearly described, this provides a fair basis for benchmarking improvements. The baselines used are relevant models for the task and datasets. Further, we had implemented additional baseline such as  FakeFlow.

---

> > > > ### Author Response · Authors · 2023-11-21
> > > >
> > > > **_Comment 10:_** **In the ablation study examining different variants of the proposed model, such as LM, AMR, LE|AMR, and LE|WikiAMR, the authors should provide significance hypothesis test results (e.g., T-test) alongside the standard deviation of the metrics across multiple runs. Including these statistical measures would enhance the interpretability of the results.**
> > > >
> > > > **_Response:_** We conducted two-tailed t-tests to evaluate the significance of differences between the accuracy of EA$^2$N variants on randomly selected samples. The hypotheses were defined as follows:
> > > >
> > > > - **Null Hypothesis (H₀)**: There is no significant difference between the accuracy of EA$^2$N variants on randomly selected samples.
> > > > - **Alternative Hypothesis (H₁)**: There is a significant difference between the accuracy of EA$^2$N variants on randomly selected samples.
> > > >
> > > > The t_statistic is calculated from the formula below:
> > > >
> > > > $$
> > > > t = \frac{\bar{x1} - \bar{x2}}{ \sqrt{\frac{s1^2}{n1} + \frac{s2^2}{n2}}}
> > > > $$
> > > >
> > > > Here $s1$ and $s2$ are the standard deviation, $n1$ and $n2$ represent the samples considered, and $\bar{x1}$ and $\bar{x2}$ represent the mean accuracy for Model 1 and Model 2, respectively.
> > > >
> > > > The t-statistic scores and p-values for all the models based on randomly selected 100 sample sets, grouped by dataset, are presented in  the given Table. Standard deviation for individual models such as LE, AMR, LE$\mid$AMR, LE$\mid$WikiAMR are 2.53, 3.21, 2.32, 2.71 for Politifact and 2.11, 3.07, 2.73, 2.35 for Gossipcop, respectively. It is evident from the table that the obtained significance values for the corresponding variants are less than 0.01, contradicting the null hypothesis. This statistical interpretation indicates a significant difference between the evaluated variants in both the Politifact and Gossipcop datasets. The same is presented in the revised manuscript (Appendix A.3.4).
> > > >
> > > > | Method                     | Politifact|        | Gossipcop|         |
> > > > |----------------------------|------------------|-----------------|-----------------|-----------------|
> > > > |                     | t-statistics| p-value  |  t-statistics|  p-value      |
> > > > | LE and AMR          | 6.19      | 3.39e-09    | 8.21|2.64e-14|
> > > > | AMR and LE$\mid$AMR        | 14.39    | 1.30e-32    | 14.46|7.96e-33|
> > > > | LE$\mid$AMR and LE$\mid$WikiAMR     | 12.40    | 1.63e-26     | 6.85|8.74e-11|
> > > >
> > > > **References**
> > > > - [1] Tetko IV. Associative neural network. Methods Mol Biol. 2008;458:185-202. doi: 10.1007/978-1-60327-101-1_10. PMID: 19065811.
> > > >
> > > > - [2, 3] AMR Parsing as Sequence-to-Graph Transduction (Zhang et al., ACL 2019)
> > > >
> > > > - [4] Nguyen Vo and Kyumin Lee. Where are the facts? searching for fact-checked information to alleviate the spread of fake news. In Bonnie Webber, Trevor Cohn, Yulan He, and Yang Liu (eds.), Proceedings of the 2020 Conference on Empirical Methods in Natural Language Processing (EMNLP), pp. 7717–7731, Online, November 2020. Association for Computational Linguistics.

---

> > > > > ### Author Response · Authors · 2023-11-22
> > > > >
> > > > > Dear Reviewer PY31,
> > > > >
> > > > > Thank you for dedicating your valuable time to review our submission. We want to gently remind you that the final discussion period is concluding shortly.  We have tried our best to address your concerns in our responses, which, hopefully, answered your questions. If you have any further concerns, please feel free to let us know.
> > > > >
> > > > > Regards,
> > > > > Authors of Submission2835

---

> > > > > > ### Comment · Reviewer_PY31 · 2023-11-23
> > > > > > **Final remarks on the authors' responses**
> > > > > >
> > > > > > I would like to thank the authors for elaborating on the various discrepancies and providing the necessary experimentation. However, I will not be changing my score. From the authors’ comments and the overall paper, it is quite explicit that the sole “novel” contribution of this work is centred around the augmentation of the WikiAMR graph. This contribution isn’t enough for a conference like ICLR. Further, even in the WikiAMR generation process, what the authors claim to be a novel algorithm is simply a variant of the A* search algorithm (which is not new). One of my major concerns (still prevailing) is the technical soundness of the Relation Encoder in the Path-aware Graph Learning Module (as I have described in my review). It raises questions about the usefulness of the graph encoder as a whole, and using the textual encoders along with the graph encoder doesn’t seem to solve this issue. The baseline comparison still remains unfair as the baselines, like GCAN, use a different set of features, which cannot be compared to the proposed architecture.

---

> ### Author Response · Authors · 2023-11-23
>
> Dear Reviewer,
>
> Thanks for your comment.
>
> Let me put this clearly:
>
> 1. Regarding the relation encoder, probably you are missing one point: an WikiAMR contains several entities. On average, a document contains more than 20 entities. The example you are trying to convey is that we reduced every path into direct paths. That is not true. If you check Table 2, it is saying we found only about 1000 paths from WikiAMR in Politifact, while the total data set contains $815\times ^{20}C_2$ (a conservative number) pairs of entities. Hence, an WikiAMR only reduces paths for a fraction of relationships. I could not understand your argument about why this is not preserving both the shortest paths found in AMR and the evidence path. I think this is our fault in presenting the text as we mentioned: "It is observed that the majority of entities in both datasets are linked using 1 hop, indicating direct relations between them." This line should read, "It is observed that the majority of entities where we found connections in both datasets are linked using 1 hop, indicating direct relations between them." We will be uploading the modified manuscript soon. Further, in support of this, we also showed through the ablation study that the incorporation is significant in terms of experimental results as well.
> 2. I think developing a new feature set is also quality research. Many of the reviewers pointed out work that is orthogonal to our problem statement to deny our contributions (by referring, they also used evidence). If you think we should remove GCAN from the comparison table, we will be happy to remove that. However, we are trying to understand that if an orthogonal problem can be a reason for denying contribution, then comparison should also be seen in the same light.
> 3. We did not try to reinvent the wheel, as A* is sufficient for the purpose of the extraction. I hope this can't be a reason to consider this pipeline trivial. While the whole pipeline we developed is a novel concept, never used for FND problems, our comments only tried to show you that more than one component we used has significant novelty. Specifically, the data structure, WikiAMR that we developed has the potential to be utilized for other NLP problems (the same is pointed out by other reviewers).
>
> Finally, we understand that the quality of ICLR is very good. Surely we did not present substandard work here. We achieved results that are greater than those of the SOTA methods with our novel pipeline and novel feature set.
>
> Thank you again for your reviews and suggestions on new experiments, which made our paper much better!
>
> Regards,

---

### Official Review · Reviewer_b3Lz · 2023-10-23

**Soundness:** 3 good
**Presentation:** 3 good
**Contribution:** 2 fair
**Rating:** 3
**Confidence:** 4

**Summary:**

This paper proposes an evidence-based AMR attention NN for fake news detection. The proposed framework encompasses a combination of language encoder and graph encoder to detect fake news using AMR and wiki external knowledge. The experimental results show the effectiveness of the proposed model.

**Strengths:**

The overall structure is well organized. The experimental results show the effectiveness of the wiki's external knowledge and AMR information. The ablation studies and case studies are reasonable.

**Weaknesses:**

The overall novelty of the method is not enough for the ICLR conference. The model is an ensemble and common usage (like transformer graph). There are some similar methods in other references. Evidence-aware Fake News Detection with Graph Neural Networks.  MUSER: A MUlti-Step Evidence Retrieval Enhancement Framework for Fake News Detection.  Detecting Out-of-Context Multimodal Misinformation with interpretable neural-symbolic model.

The ablation studies should add the ELF and CLF-based experiments to show the effectiveness.

It's better to add more datasets about fake news detection, such as  Snop.

The format of reference should be revised (Devlin et al., 2019)

**Questions:**

See aboove.

---

> ### Author Response · Authors · 2023-11-21
>
> We thank the reviewer for the detailed review. Below, we address all the raised concerns and questions.
>
> **_Comment 1.1:_** **The overall novelty of the method is not enough for the ICLR conference. The model is an ensemble and common usage (like transformer graph).**
>
> **_Response:_** While our work amalgamates some existing architectures, we would like to emphasize the unique contributions and novel analyses embedded within our study.
>
> - Central to our contribution is the introduction of a novel evidence-based structure called WikiAMR. WikiAMR is substantially different than AMR as it contains circumstantial linking between entities through undirected edges. While latter is a DAG graph the former is not. WikiAMR intricately captures the comprehensive semantic structure of news articles along with evidence linking. One may observe that from the ablation study (Section 6.2 of the paper) this additional links provide significant (p-value less than 0.01) improvement in the model’s performance.
>
> - WikiAMR, with its intricate structure and nuanced representation of semantic relationships, closely parallels the **associative network [1]** observed in the human brain. By leveraging these principles, WikiAMR adeptly correlates and links diverse entities within news articles based on both actual evidence and circumstantial cues. This dual emphasis on concrete evidence and associative reasoning aligns with the human brain's cognitive processing. This enables the model to foster a more holistic comprehension of news articles.
>
> - Further, we propose an algorithm to generate WikiAMR efficiently. This involves CLF and ELF where we introduced two cost functions that effectively reduces the search space. Finally, we judicially concatenate features from WikiAMR graph into the language encoder in the context of FND.
>
> **_Comment 1.2:_** **There are some similar methods in other references. Evidence-aware Fake News Detection with Graph Neural Networks. MUSER: A MUlti-Step Evidence Retrieval Enhancement Framework for Fake News Detection. Detecting Out-of-Context Multimodal Misinformation with interpretable neural-symbolic model.**
>
> **_Response:_** Although, there might be some correspondence present between our approach but the below highlights some key differences:
>
> - *Evidence-aware Fake News Detection with Graph Neural Networks* (GET) aligns with aspects of KAN in its utilization of graph structures for fake news detection, but it adopts neural networks instead of transformers. Both GET and KAN leverage graph-based methodologies, focusing on capturing structural dependencies and semantic representations within textual data without an optimized framework. However, our model diverges significantly by employing Abstract Meaning Representation (AMR) for semantic understanding, allowing for a deeper and more nuanced representation of text compared to the graph-based semantics utilized in GET. Additionally, the results from the benchmarks presented shows that our model is much better due to the incorporation of A* algorithm to intelligently link entities within the AMR representation, facilitating a more precise understanding of textual relationships.
> - Our AMR-based model stands apart from MUSER by preserving semantic nuances through AMR representation, contrasting MUSER's text-centric summarization. While MUSER relies on multi-step retrieval, our model employs an A* algorithm within AMR to intelligently link entities, capturing nuanced relationships more directly and efficiently. Additionally, our approach showcases superior performance, leveraging AMR's semantic fidelity for fake news detection, surpassing MUSER's text-only reliance. The A* algorithm within AMR enhances our model's ability to comprehend intricate textual dependencies, distinguishing it significantly from MUSER's methodology.
> - For the last paper, our AMR-based model focuses on semantic nuances within text using AMR and the A* algorithm, emphasizing textual understanding for fake news detection. In contrast, the model detecting out-of-context multimodal misinformation specializes in identifying discrepancies between visual and textual data, operating within a multimodal domain to uncover context misalignments. The domain thus between the tasks is different for both of these models and architecturally different based on the multi-modal handling.

---

> > ### Author Response · Authors · 2023-11-21
> >
> > **_Comment 2:_** **The ablation studies should add the ELF and CLF-based experiments to show the effectiveness.**
> >
> > **_Response:_** Based on your suggestion, we have performed a sensitivity analysis to examine the influence of different ELF (Entity Level Filtering) and CLF (Context Level Filtering) thresholds on model's performance. The average entity values for ELF thresholds ($\gamma$) and CLF thresholds ($\delta$), based on 100 samples, are displayed in the Figure 7, Appendix A.3.5 .
> >
> > The threshold values significantly impact system behavior. Higher thresholds lead to fewer entities, indicating stricter linking and classification criteria. When entities decrease, the graph tends to converge from WikiAMR to AMR, potentially reducing accuracy. Conversely, a higher entity count results in a larger graph size, escalating training times. A $\gamma$ of 0.4 maintains a balance between accurate and manageable linking of primary entities. Meanwhile, a $\delta$ of 0.1 helps control relevancy of the linked paths without compromising accuracy significantly. Detailed description is added in section Appendix A.3.5 in the revised manuscript.
> >
> >
> >
> > **_Comment 3:_** **It's better to add more datasets about fake news detection, such as Snop.**
> >
> > **_Response:_** In order to evaluate the generalizability of our model, we additionally conducted tests on the publicly available Snopes dataset [2], consisting of a total of 1703 fact-checking articles covering various political topics sourced from the fact-checking website [snopes.com](http://snopes.com/). This dataset encompasses multiple classes, including false, true, mostly false, mostly true, scam, unknown, etc. Given that EA$^2$N is a fake/real classification model, we focused exclusively on the true and false classes. For the final assessment, we utilized 1106 articles classified as fake and 182 articles classified as true. To address class imbalance in the dataset, we employed standard NLP based data augmentation techniques. The results presented in below table clearly demonstrate that our model EA$^2$N surpasses other models by a significant margin. We have compared our model EA$^2$N (LE$\mid$WikiAMR) with 1) FakeFlow (FF) 2) Language Encoder (LE) 3) AMR 4) Language Encoder with AMR (LE$\mid$AMR). It is evident form the results that when we integrate evidence in AMR into our final model (LE$\mid$WikiAMR), there is 2-5\%, 1-6\% gain in accuracy and F1- score respectively from all the other models. The same is incorporated in the revised manuscript (Appendix A.3.3).
> >
> > | Method                     | F1         | ACC           |
> > |----------------------------|------------------|-----------------|
> > | FF            |0 .8712      | 0.8542    |
> > | LE          | 0.8620     | 0.8511    |
> > | AMR       | 0.8961     | 0.8769     |
> > | LE$\mid$AMR        | 0.9144      | 0.8869     |
> > | LE$\mid$WikiAMR (EA$^2$N)       |  **0.9212**      | **0.9045**     |
> >
> > **_Comment 4:_** **The format of reference should be revised (Devlin et al., 2019)**
> >
> > **_Response:_** In the revised version of the manuscript, we have corrected the citation format.
> >
> > **References**
> > - [1] Tetko IV. Associative neural network. Methods Mol Biol. 2008;458:185-202. doi: 10.1007/978-1-60327-101-1_10. PMID: 19065811.
> > - [2] Nguyen Vo and Kyumin Lee. Where are the facts? searching for fact-checked information to alleviate the spread of fake news. In Bonnie Webber, Trevor Cohn, Yulan He, and Yang Liu (eds.), Proceedings of the 2020 Conference on Empirical Methods in Natural Language Processing (EMNLP), pp. 7717–7731, Online, November 2020. Association for Computational Linguistics.

---

> > > ### Author Response · Authors · 2023-11-22
> > >
> > > Dear Reviewer  b3Lz,
> > >
> > > Thank you for dedicating your valuable time to review our submission. We want to gently remind you that the final discussion period is concluding shortly.  We have tried our best to address your concerns in our responses, which, hopefully, answered your questions. If you have any further concerns, please feel free to let us know.
> > >
> > > Regards,
> > > Authors of Submission2835

---

### Official Review · Reviewer_hTw2 · 2023-10-31

**Soundness:** 3 good
**Presentation:** 3 good
**Contribution:** 2 fair
**Rating:** 5
**Confidence:** 2

**Summary:**

This paper introduces EA2N, an Evidence-based AMR Attention Network for Fake News Detection. The proposed framework leverages Abstract Meaning Representation (AMR) and incorporates knowledge from Wikidata to detect fake news. It combines language encoder and graph encoder to effectively capture complex semantic relations and improve the reliability of incorporating external knowledge. Their experiments demonstrate the effectiveness of EA2N compared to state-of-the-art methodologies.

**Strengths:**

- The paper is well-written and easy to follow.
- The authors claim they will release the code once the discussion forum start.
- Compared with the baselines used in this paper, EA2N achive effective resuls.

**Weaknesses:**

- The idea of use external knowledge to enhane fake news detection is not new.

**Questions:**

No

---

> ### Author Response · Authors · 2023-11-20
>
> We thank the reviewer for the positive feedback and would like to further clarify some points.
>
> **_Comment:_** **The idea of use external knowledge to enhance fake news detection is not new.**
>
> **_Response:_**  The proposed method uses external knowledge for fake news detection. However, the methodology used to collect evidence from external sources is novel. In this, we propose a new representation of AMR called as WikiAMR, which links the facts among the entities. WikiAMR comprises interconnected undirected evidence paths between entities present in the original AMR, which helps to prioritize evidence over the claims by the path-aware learning module encountered in the news document. Hence, our contribution is twofold: first, we develop a WikiAMR graph containing evidence among entities, and the other is to use it via path-aware graph learning model for FND that greatly impacts models interpretability and its architecture.

---

> > ### Comment · Reviewer_hTw2 · 2023-11-22
> >
> > I am convinced by other reviewers' comments, so I change my score from 6 to 5.

---

> > > ### Author Response · Authors · 2023-11-22
> > >
> > > Dear Reviewer,
> > >
> > > We have already addressed the concerns of other reviewers. Hence, we are requesting you to review our responses as well as the revised manuscript.
> > >
> > > Thank you!
> > >
> > > Regards,

---

### Official Review · Reviewer_cpwY · 2023-10-31

**Soundness:** 3 good
**Presentation:** 3 good
**Contribution:** 2 fair
**Rating:** 5
**Confidence:** 4

**Summary:**

This paper focuses on the detection of fake news on social media through the integration of external knowledge and evidence. The authors argue that existing related papers encounter three primary challenges: (1) the difficulty of capturing long-term and intricate semantic relationships, (2) unreliability and time-consuming knowledge acquisition processes, and (3) reliance on potentially unreliable information sourced from social media users. To address these challenges, the authors introduce a novel model, termed the Evidence-based AMR Attention Network (EA$^2$N). This model incorporates an Abstract Meaning Representation (AMR) graph and a refined knowledge graph derived from Wikidata to extract evidential features, while employing BERT to capture semantic features. Subsequently, these two sets of features are concatenated to predict the veracity labels. Experimental validation is conducted to demonstrate the model's effectiveness.

**Strengths:**

1. The paper focuses on a practical and challenging issue, fake news detection based on external evidence.
2. The model is the first attempt to utilize an AMR graph to enhance the detection of fake news.
3. The experiments are extensive, and significantly and consistently outperform the state-of-the-art model, which can prove the effectiveness of the proposed model.

**Weaknesses:**

However, despite the superior performance of the proposed model, there still exist some weaknesses in the paper.
1. In the Introduction section, the authors summarize several problems in existing fake news detection (FND) works. However, the author seems not to have successfully solved all the problems.
2. AMR parser has been a well-studied technique used by a variety of NLP tasks. Therefore, the novelty of the idea of incorporating AMR into FND is limited.
3. This paper proposes a FND model that uses AMR and Wikidata knowledge. However, this method is not only suitable for the FND task, but can also be applied to other knowledge-rich NLP tasks, e.g. sentiment analysis and intent detection. So why do the authors only focus on the fake news detection task? In other words, which characteristics of EA$^2$N determine that this method is only suitable for fake news detection?
4. The sensitivity analysis of the thresholds $\gamma$ and $\delta$ should be provided.

**Other details:**
1. In order to ensure standardization, citations should be revised, e.g. Brewer et al. (2013) -> (Brewer et al. 2013).
2. The algorithm in Sec.3.3.1 involves being converted into a figure or a standard algorithm table for ease of understanding.

**Questions:**

1. AMR is a kind of graph to capture semantic correlations of documents. And dependency tree can play the same role with AMR. Therefore, what are the advantages of AMR compared to dependency trees?
2. As discussed in the Weakness part, whether this paper solves the problem *"the way of incorporating external knowledge into these models is not highly reliable and time-consuming."*

---

> ### Author Response · Authors · 2023-11-20
>
> We thank to the reviewer for providing valuable questions and concerns. Justification of each point is addressed below.
>
> **_Comment 1:_** **In the Introduction section, the authors summarize several problems in existing fake news detection (FND) works. However, the author seems not to have successfully solved all the problems.**
>
> **_Response:_** In the introduction section we have discussed about the following problems:
>
> *Problem 1- SOTA models struggle to maintain longer text dependencies and less effective to capture complex semantic relations such as events, locations, trigger words and so on.*
>
> - We are using Abstract Meaning Representation, which captures the complex semantic relations for long sentences using Sembanking language. AMR provides a structured and abstract representation of the underlying semantics of a sentence. This structure can assist in capturing complex relationships between entities, events, and other elements in a sentence. SOTA models, especially those based on neural networks, often benefit from structured representations that help them generalize better across different examples. AMR is designed to explicitly represent the meaning of a sentence, including events, entities, and their relationships. This can be particularly valuable in tasks like Fake News Detection.
>
> *Problem 2-  The way of incorporating external knowledge into these models is not highly reliable and time consuming. For example, KAN [1] only considers single entity contexts and fail to link context between two entities. On the other hand, FinerFact [2] gather supported claims from social platforms that is time-consuming. Although social authenticity produces good results, these information can be manipulated by social media users for personal gain.*
>
> - The proposed method provides higher reliability through the proposed evidence path extracted from WikiData over the entities in the AMR graph. This also provides an explainable way to access the facts. Linking multiple entities was avoided by earlier work as that may required higher execution time. We have dealt with this by efficient A* search using ELF and CLF to integrate the path. In other words, our proposed method provides better reliability with reasonable execution time.
>
>
> **_Comment 2:_** **AMR parser has been a well-studied technique used by a variety of NLP tasks. Therefore, the novelty of the idea of incorporating AMR into FND is limited.**
>
> **_Response:_** We agreed that AMR had been explored in many NLP domains with its capability of representing complex semantic relations present in text using a graphical representation. We propose a new representation of AMR called WikiAMR, which links the facts among the entities. WikiAMR comprises interconnected undirected evidence paths between entities present in the original AMR, which helps to prioritize evidence over the claims by the path-aware learning module encountered in the news document. Hence, our contribution is twofold, first, we develop a WikiAMR graph containing evidence among entities, and the other is to use it via path-aware graph learning model for FND that greatly impacts models interpretability and its architecture.
>
>
> **_Comment 3:_** **This paper proposes a FND model that uses AMR and Wikidata knowledge. However, this method is not only suitable for the FND task, but can also be applied to other knowledge-rich NLP tasks, e.g. sentiment analysis and intent detection. So why do the authors only focus on the fake news detection task? In other words, which characteristics of EAN determine that this method is only suitable for fake news detection?**
>
> **_Response:_** Path-aware graph learning with WikiAMR proposed here may be useful for other NLP tasks. However, the detection of misinformation is nothing but verification of content using evidence. While working on FND problem, we focused on evidence linking. AMR, here, provides the semantic representation that helps in finding evidence from Wiki Knowledge.
>
> **_Comment 4:_** **The sensitivity analysis of the thresholds $\gamma$ and $\delta$ should be provided.?**
>
> **_Response:_** We have performed a sensitivity analysis to examine the influence of different ELF (Entity Level Filtering) and CLF (Context Level Filtering) thresholds on model's performance. The average values for ELF thresholds (γ) and CLF thresholds (δ), based on 100 samples are reported.
>
> The threshold values significantly impact system behavior. Higher thresholds lead to fewer entities, indicating stricter linking and classification criteria. When entities decrease, the graph tends to converge from WikiAMR to AMR, potentially reducing accuracy. Conversely, a higher entity count results in a larger graph size, escalating training times.  A $\gamma$ of 0.4 maintains a balance between accurate and manageable linking of primary entities. Meanwhile, a $\delta$ of 0.1 helps control relevancy of the linked paths without compromising accuracy significantly. (Figure 7, Appendix A.3.5)

---

> ### Author Response · Authors · 2023-11-20
>
> **_Comment 5:_** **In order to ensure standardization, citations should be revised, e.g. Brewer et al. (2013) -> (Brewer et al. 2013).**
>
> **_Response:_** In the revised version of the manuscript, we have corrected the citation format.
>
> **_Comment 6:_** **The algorithm in Sec.3.3.1 involves being converted into a figure or a standard algorithm table for ease of understanding.**
>
> **_Response:_** Section 3.3.1 contains an example of AMR, not an algorithm. It would be good if you can clarify the question.
>
> **_Comment 7:_** **AMR is a kind of graph to capture semantic correlations of documents. And dependency tree can play the same role with AMR. Therefore, what are the advantages of AMR compared to dependency trees?**
>
> **_Response:_** In the context of fake news detection, Abstract Meaning Representation (AMR) and dependency trees can play distinct roles, and there are specific advantages to using AMR over dependency trees. AMR holds the deeper semantic structure of sentences that helps in the detection of false information that may not be evident in the syntactic or lexical structure alone. Other advantages such as, it helps to uncover instances where the intended meaning diverges from the apparent surface structure, making it more adept at identifying subtle manipulations within news content. On the other hand, the dependency tree fails to do so with only lexical understanding.
>
> **_Comment 8:_** **As discussed in the Weakness part, whether this paper solves the problem "the way of incorporating external knowledge into these models is not highly reliable and time-consuming."**
>
> **_Response:_**  As discussed in the above comments, the proposed method provides higher reliability with reasonable time. Further, this method is probabilistically better because of existing facts instead of social information.
>
> **References**
> - [1] Yaqian Dun, Kefei Tu, Chen Chen, Chunyan Hou, and Xiaojie Yuan. Kan: Knowledge-aware attention network for fake news detection. AAAI, 35(1):81–89, May 2021
> - [2] Yiqiao Jin, Xiting Wang, Ruichao Yang, Yizhou Sun, Wei Wang, Hao Liao, and Xing Xie. Towards fine-grained reasoning for fake news detection. AAAI, 36:5746–5754, 06 2022

---

> > ### Author Response · Authors · 2023-11-22
> >
> > Dear Reviewer cpwY,
> >
> > Thank you for dedicating your valuable time to review our submission. We want to gently remind you that the final discussion period is concluding shortly.  We have tried our best to address your concerns in our responses, which, hopefully, answered your questions. If you have any further concerns, please feel free to let us know.
> >
> > Regards,
> >
> > Authors of Submission2835

---

### Author Response · Authors · 2023-11-20

We appreciate the insightful and constructive feedback provided from the reviewers. Their valuable suggestions have played a crucial role in enhancing our manuscript, strengthening its contributions to the field. We have incorporated below mentioned changes in the revised manuscript and highlighted with red color:

- **Explanation of cost function in A$^\*$ algorithm:** We have improved the explanation of the cost function within the A$^\*$ algorithm, addressing specific points highlighted by the reviewers. This enhancement aims to provide a clearer and more comprehensive understanding of the evidence linking algorithm. (In Section 3.3.2)

- **Experiment on additional dataset:** In order to evaluate the generalizability of the model, we have conducted a comparative study on publicly available dataset (snopes). This addition extends the scope of our study and strengthens the empirical foundation of our findings. (In Appendix A.3.3)

- **T-Tests for EA2N variants**:  In line with reviewers feedback, we have performed t-tests for different variants of EA$^2$N. This statistical analysis provides a robust evaluation framework, offering insights into the significance of differences between model variations. (In Appendix A.3.4)

- **Sensitivity analysis**: We have conducted a detailed sensitivity analysis of the $\gamma$ and $\delta$ parameters, shedding light on their impact on our proposed approach. This analysis contributes to a better understanding of the evidence linking in AMR using ELF and CLF.  (In Appendix A.3.5)

---

### Meta-Review · Area_Chair_h7me · 2023-12-05

**Metareview:**

The paper proposes an approach to fake news detection that is based on the integration of external knowledge and evidence. Specifically, the authors propose EA2N, an Evidence-based AMR Attention Network that utilises  external knowledge from Wikidata within an Abstract Meaning Representation (AMR) graph. This is the first paper to apply an AMR graph approaach to the task of fake news. The authors experimentally demonstrate that their approach outperforms the current sota. One of the reviewers’ key issue is that of technical novelty, as they explain (among others) that AMR has already been explored for a number of NLP tasks, and the current paper seems to “combine several pre-existing architectures”. While the type of technical novelty in this paper can be considered incremental, the authors have nevertheless presented a novel application that includes an adaptation of existing methods to fit the task of fake news detection, and experimentally demonstrate that their approach improves the current sota. The main issue I would think, as presented in detail by reviewer PY31, is the question about the technical soundness of the Relation Encoder in the Path-aware Graph Learning Module. The reviewer has read the authors’ response on this matter but they remain unconvinced. Further elaboration and explanation is needed on this matter so the reader can fully appreciate the significance of this.

Overall, the paper needs thorough proofreading (as mentioned by one of the reviewers too), and there are already a few grammar errors in the abstract.

The authors should also ensure they incorporate all of the reviewers' feedback and new experiments in their paper as they will substantially improve quality.

**Justification For Why Not Higher Score:**

The paper has an average score of 4, and none of the reviewers are enthusiastic about it. A key weakness they see relates to lack of technical novelty.

**Justification For Why Not Lower Score:**

I would still think that the paper presents a novel application of existing methods that is shown to consistently improve the current sota.

---

### Decision · Program_Chairs · 2024-01-16

Reject